# Altered basal ganglia output during self-restraint

**Bon-Mi Gu[1,2]\*, Joshua D Berke[1,3]**

[1]Department of Neurology, University of California, San Francisco, San Francisco, United States; [2]Department of Neurology and Neurological Sciences, Stanford University, Stanford, United States; [3]Department of Psychiatry and Behavioral Sciences, Neuroscience Graduate Program, Kavli Institute for Fundamental Neuroscience, Weill Institute for Neurosciences, University of California, San Francisco, San Francisco, United States

**Abstract** Suppressing actions is essential for flexible behavior. Multiple neural circuits involved in behavioral inhibition converge upon a key basal ganglia output nucleus, the substantia nigra pars reticulata (SNr). To examine how changes in basal ganglia output contribute to self-restraint, we recorded SNr neurons during a proactive behavioral inhibition task. Rats responded to *Go!* cues with rapid leftward or rightward movements, but also prepared to cancel one of these movement directions on trials when a *Stop!* cue might occur. This action restraint – visible as direction-selective slowing of reaction times – altered both rates and patterns of SNr spiking. Overall firing rate was elevated before the *Go!* cue, and this effect was driven by a subpopulation of direction-selective SNr neurons. In neural state space, this corresponded to a shift away from the restrained movement. SNr neurons also showed more variable inter-spike intervals during proactive inhibition. This corresponded to more variable state-space trajectories, which may slow reaction times via reduced preparation to move. These findings open new perspectives on how basal ganglia dynamics contribute to movement preparation and cognitive control.

\*For correspondence:
bonmigu@stanford.edu

Competing interest: The authors declare that no competing interests exist.

## Editor's evaluation

The article provides an interesting and timely insight into the role of the basal ganglia output in proactive inhibition. By examining single-cell responses as well as population activity, the authors establish neural signals of behavioral control and show that animals' outcome history influences both firing rates and variability of basal ganglia output activity.

## Introduction

Fluid, efficient behavior often involves simply triggering well-learned behaviors. However, flexibility requires that such behaviors can be suppressed, should circumstances change. This capacity for behavioral inhibition is considered central to cognitive control (*Bari and Robbins, 2013*), and is compromised in a range of neurological and psychiatric disorders (*Chambers et al., 2009*).

Behavioral inhibition can be 'reactive' – for example, promptly responding to an unexpected *Stop!* cue by cancelling upcoming actions. Reactive inhibition has been shown to involve fast cue responses in frontal cortex and basal ganglia pathways (*Jahanshahi et al., 2015*; *Wager et al., 2005*), including from the subthalamic nucleus (STN) to SNr (*Schmidt et al., 2013*). This rapid response to stimuli can transiently, and broadly, retard action initiation, providing time for a second set of basal ganglia mechanisms to cancel actions (*Mallet et al., 2016*; *Schmidt and Berke, 2017*).

By contrast, 'proactive' inhibition refers to an altered state of preparation, in which particular actions are restrained (*Cai et al., 2011*; *Claffey et al., 2010*). Proactive inhibition has been argued to be especially important for human life (*Aron, 2011*), and is behaviorally apparent as longer reaction times (RTs) selectively for the restrained action. The underlying mechanisms are not well understood, but have been proposed (*Aron, 2011*) to involve the pathway from striatum 'indirectly' to SNr, via globus pallidus pars externa (GPe). In a prior study (*Gu et al., 2020*) we therefore recorded from GPe neurons as rats engaged proactive inhibition toward a specific action. The state of being prepared to stop did not involve an overall net change in GPe firing rate, but rather a shift at the level of neural population dynamics away from action initiation, and toward the alternative action. One objective of the present work was to determine whether a corresponding preparatory change in population dynamics is visible 'downstream' in SNr, thereby altering basal ganglia output.

Basal ganglia output neurons are thought to affect behavior not just via their firing rates, but also via their firing patterns and synchrony (*Rubin et al., 2012*). In particular, Parkinson's disease (PD) is associated with an increase in firing variability and synchronous bursting, often without rate changes (*Lobb, 2014*; *Willard et al., 2019*). As PD is characterized by slowed movement initiation (*Low et al., 2002*), we assessed whether related physiological changes are present when movements are slowed as the result of proactive inhibition.

## Results

### RTs are selectively slowed with proactive inhibition

We used a selective proactive inhibition task (*Gu et al., 2020*), a variant of our extensively characterized rat stop-signal task (*Leventhal et al., 2012*; *Schmidt et al., 2013*; *Mallet et al., 2016*; *Figure 1A*). Rats start a trial by nose-poking an illuminated start port. To proceed, they need to maintain this position for a variable delay (500–1250 ms, uniform distribution) until a *Go!* cue is presented (1 or 4 kHz tone, indicating a leftward or rightward movement, respectively). If the movement is initiated rapidly after the *Go!* cue (RT limit <800 ms), and completed correctly and promptly (movement time limit, MT <500 ms), rats are rewarded with a sugar pellet dispensed at a separate food hopper. On a subset of trials, the *Go!* cue is followed by a *Stop!* cue (white noise burst; delay after *Go!* cue onset = 100–250 ms). This indicates that the rat should not initiate a movement, and instead hold their nose in the Center port for at least 800 ms (after *Go!* cue onset) to trigger reward delivery.

To probe selective proactive inhibition, the three possible start ports were associated with different *Stop!* cue probabilities (counterbalanced across rats; *Gu et al., 2020*, *Figure 1B*, *Table 1*). These were: no possibility of *Stop!* cue ('No-Stop'); 50% probability that a left *Go!* cue will be followed by the *Stop!* cue ('Maybe-Stop-left'); and 50% probability that a right *Go!* cue will be followed by the *Stop!* cue ('Maybe-Stop-right'). We obtained SNr microelectrode recordings from rats (n=10) that had successfully learned this proactive task, as indicated by significant and selective slowing of RTs for movements contraversive to the implant side (*Figure 1C*). For example, if electrodes were placed in the right SNr, contraversive proactive inhibition would mean longer RTs for Maybe-Stop-left trials, compared to No-Stop trials. In the same rats we compared SNr activity in sessions in which this proactive effect was significant (n=60), to sessions in which it was not (n=25; *Figure 1C*).

The sessions with significant proactive inhibition effect show RT slowing selectively for the Maybe-Stop direction (Wilcoxon signed rank tests on median RTs of Maybe-Stop-Contra versus No-Stop: Contra cues: z=6.8, p=1.1 × 10$^{-11}$), but not for the other direction (Ipsi cues: z=–1.2, p=0.22) (*Figure 1D*). Additionally, on Maybe-Stop-Contra trials rats were more likely to fail to respond quickly enough (RT limit errors; Wilcoxon signed rank tests, z=6.4, p=1.5 × 10$^{-10}$) and to select the wrong choice (i.e. not matching the *Go!* cue; Wilcoxon signed rank tests, z=6.5, p=1.1 × 10$^{-10}$).

### Selective proactive inhibition recruits specific SNr subpopulations

We examined the activity of individual SNr neurons (n=446; mean firing rate = 38 Hz, locations are shown in *Figure 2—figure supplement 1*) recorded during the sessions with significant proactive inhibition. We first compared overall cell activity between Maybe-Stop-Contra and No-Stop trials, within the same sessions. We focused on the epoch just before the *Go!* cue, as we presume that this time is critical for being 'prepared-to-stop'. Since this time epoch is before any Stop cue could occur, we included Maybe-Stop trials regardless of whether a Stop cue was subsequently presented

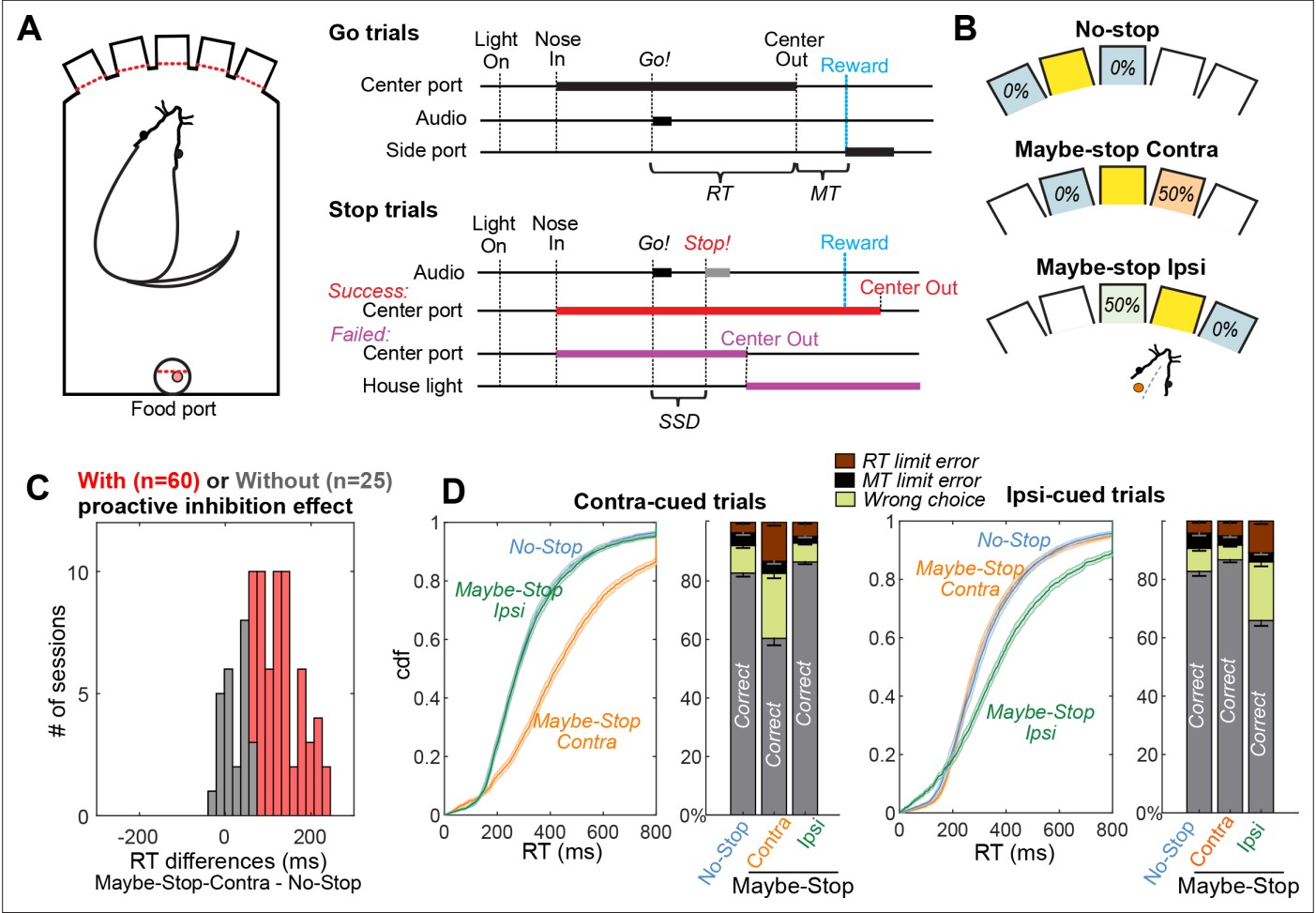

**Figure 1.** Selective proactive inhibition. (**A**) Left, operant box configuration, with dashed red lines indicating photobeams for nose detection; right, event sequence for Go and Stop trials. RT, reaction time; MT, movement time; SSD, stop-signal delay; Reward, delivery of a sugar pellet to the food port. (**B**) Trial start location indicates stop probabilities. In this example configuration with left SNr recording, illumination of the middle hole indicates that *Go!* cues instructing rightward movement may be followed by a *Stop!* cue, but *Go!* cues instructing leftward movements will not ('Maybe-Stop-Contra'). (**C**) Overall, the sessions in which SNr units were successfully recorded showed strong proactive inhibition effect (n=85, Wilcoxon signed rank tests on median RT differences between Maybe-Stop-Contra and No-Stop-Contra conditions, p=8.2 × 10⁻¹⁵). Among them, the individual sessions are considered to show proactive inhibition (red) if the reaction time difference between Maybe-Stop-Contra and No-Stop trials is statistically significant (one-tail Wilcoxon rank sum test, p<0.05). (**D**) Cumulative distribution functions (cdf) of RTs of Maybe-Stop-Contra condition show selective slowing for the contra-cued trials, but not for the ipsi-cued trials. Response ratios also show selective increase of wrong choice and RT limit errors for the Maybe-Stop direction. Shaded band and error bars, SEM across n=60 sessions with proactive inhibition effect. RT limit error = nose remained in Center port for >800 ms after *Go!* cue onset; MT limit error = movement time between Center Out and Side port entry >500 ms.

or not. Average firing rates were significantly higher in the Maybe-Stop-Contra condition (*Figure 2A*, top), and this difference was generated by a significant fraction of SNr neurons (*Figure 2A*, bottom). Example cells show significant differences before the *Go!* cue between conditions (*Figure 2B*). This elevated SNr firing with action restraint was not observed in the sessions without significant behavioral evidence for proactive inhibition (*Figure 2A*, right; 173 neurons, mean FR = 44 Hz).

A common categorization of SNr neurons distinguishes those that increase, versus decrease, firing rate in conjunction with behavioral events (e.g. *Bryden et al., 2011*; *Fan et al., 2012*; *Gulley et al., 2002*; *Sato and Hikosaka, 2002*). Decreases in firing shortly before movement onset are thought to enable movements, by disinhibiting downstream structures including the superior colliculus (*Hikosaka and Wurtz, 1983b*). If 'decrease-type' neurons are receiving more excitation during proactive inhibition, this might delay their decrease to the level needed to release movements, resulting in longer RTs. We therefore hypothesized that proactive inhibition is associated with elevated firing specifically of decrease-type neurons.

**Table 1.** Information on individual rats.

| RAT | Port location with stop probability (port1, port2 port3) | # of sessions (with/ without proactive inhibition effect) | # of cells (from sessions with/ without proactive inhibition effect) | Sessions with contralateral proactive inhibition effect (Mean ± SD) | | | |
|---|---|---|---|---|---|---|---|
| | | | | # of trials per session (Maybe-Stop-Contra/ No-Stop conditions)* | Contralateral RTs (Maybe-Stop-Contra/ No-Stop conditions, ms) | Ipsilateral RTs (Maybe-Stop-Contra/ No-Stop conditions, ms) | Stop success rates (%) |
| 1015 | 0, L50, R50% | 7/1 | 184/20 | 104±27/103±12 | 563±39/363±35 | 428±45/415±42 | 55.3±10.3 |
| 1019 | L50, 0, R50% | 5/2 | 6/4 | 110±33/122±25 | 369±47/275±30 | 271±30/321±11 | 47.8±10.9 |
| 1042 | L50, 0, R50% | 2/0 | 3/0 | 94±5/88±10 | 399±27/293±10 | 277±4/377±37 | 81.1±5.5 |
| 1043 | L50, R50, 0% | 8/1 | 30/5 | 124±25/131±35 | 474±49/353±48 | 384±51/38747 | 70.5±5.6 |
| 1063 | L50, R50, 0% | 8/4 | 21/15 | 124±31/110±25 | 347±57/226±29 | 266±49/191±18 | 26.2±5.6 |
| 1064 | L50, 0, R50% | 6/4 | 23/26 | 121±20/123±18 | 446±34/338±35 | 303±13/309±20 | 57.3±4.8 |
| 1098 | 0, L50, R50% | 5/2 | 5/2 | 102±18/109±27 | 456±17/330±38 | 324±21/327±54 | 41.7±3.1 |
| 1202 | 0, R50, L50% | 10/1 | 114/12 | 107±23/118±32 | 494±49/328±36 | 359±38/356±28 | 56.3±10.6 |
| 1296 | R50, 0, L50% | 7/2 | 53/11 | 101±17/92±11 | 435±39/330±38 | 344±41/345±28 | 61.5±13.0 |
| 1328 | R50, L50, 0% | 2/8 | 7/78 | 127±2/147±19 | 370±13/298±36 | 275±18/368±65 | 51.8±2.5 |

*Number of trials includes only those trials in which the Go! cue was presented (i.e. trials with premature center out are excluded).

We categorized cells as increase-type or decrease-type based on their change in firing rate during the 200 ms preceding movement onset (*Figure 2C*, *Figure 2—figure supplement 3*). Increase-type were more numerous, as previously reported (*Bryden et al., 2011*; *Joshua et al., 2009*). Contrary to our hypothesis, both increase-type and decrease-type neurons contributed to elevated SNr activity with action restraint (*Figure 2D*, left). There was no relationship between the extent of elevated firing in Maybe-Stop trials, and the firing rate change before movement onset (*Figure 2D*, right).

We then examined whether proactive inhibition effects were related to neurons' response selectivity, operationally defined as the firing rate difference between Contra and Ipsi actions just before Center Out (*Figure 2E*, inset). We found a significant relationship: neurons more active just before Ipsi compared to just before Contra movements ('Ipsi>Contra') showed elevated firing when Contra actions might need to be cancelled (*Figure 2F and G*). No such relationship was found for neurons with the opposite selectivity ('Contra>Ipsi'; *Figure 2F and G*).

We next considered the interaction between direction selectivity and increases versus decreases in firing, in proactive inhibition. A cell could be classified as 'Ipsi>Contra' because it preferentially increases firing with Ipsi movements, or because it preferentially decreases firing with Contra movements. We found that both subtypes had elevated firing before the *Go!* cue on Maybe-Stop-Contra trials (*Figure 2—figure supplement 2*). Therefore, the proactive effect was not simply a matter of SNr cells that pause with Contra movements starting from a higher baseline rate, though this may contribute.

## Restraining one action biases population dynamics toward the alternative action

The elevated average firing rate of Ipsi>Contra cells suggests a preparatory bias toward Ipsi action, at times when Contra actions might need to be cancelled. To examine this further we turned to a state-space analysis. This allows us to examine the effects of proactive inhibition at the level of SNr neural populations, and compare to our prior results for GPe populations (*Gu et al., 2020*). We extracted principal components from the average firing rates of each neuron during Contra and Ipsi movements (*Figure 3—figure supplement 1*) and used these to visualize neural population trajectories (*Figure 3A*, top). We wished to assess a potential bias toward movement initiation in general, rather than withholding action ('Initiation') and separately assess bias toward one specific action versus the other ('Selection'). We therefore defined 'Initiation' and 'Selection' axes using the common, and distinct, aspects of the neural trajectories respectively, during the 200 ms before action initiation. Specifically, the Initiation Axis is a line drawn between the average state-space positions at -200 and 0

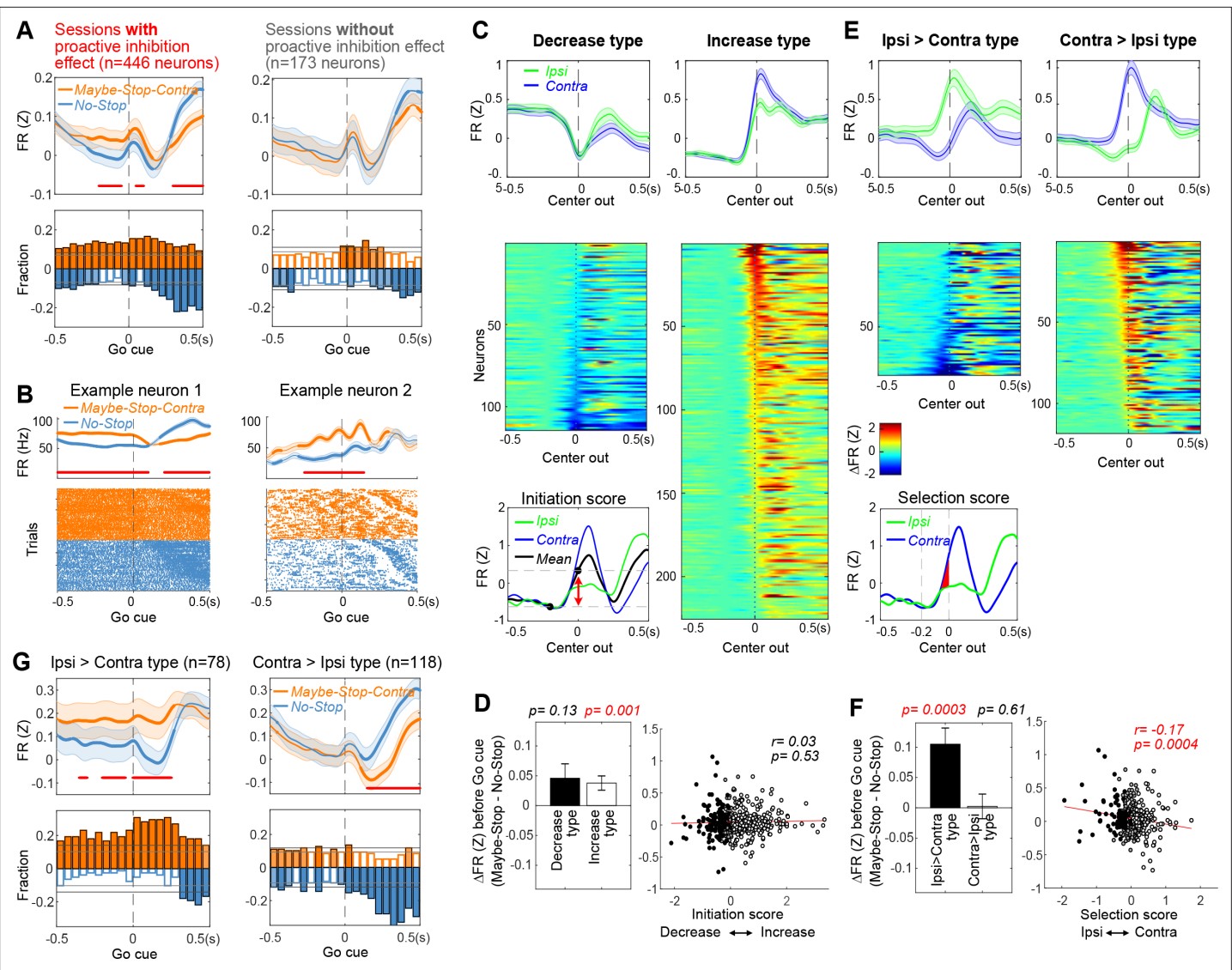

**Figure 2.** Elevated firing rates of specific substantia nigra pars reticulata (SNr) subpopulations with selective proactive inhibition. (**A**) Top left: before the *Go!* cue SNr firing is elevated in Maybe-Stop-Contra compared to No-Stop conditions. This occurs selectively in sessions with behavioral evidence of proactive inhibition (left). Each neuron's firing rate is Z-scored and averaged (over all trials in which a *Go!* cue was presented, regardless of *Stop!* cues). Shaded band, ± SEM across n=446 (left) or n=173 neurons (right). Thicker lines indicate significant differences between conditions (p<0.05; Wilcoxon signed rank tests at each time point) and red lines at the bottom indicate times with significant difference remaining after Bonferroni correction (by the number of 50 ms time bins; p<0.05). Bottom left: fraction of SNr neurons whose firing rate significantly differs between conditions, across time (p<0.05; Wilcoxon rank sum tests in each 50 ms bin). Higher firing with Maybe-Stop-Contra, No-Stop conditions are shown as positive (orange) or negative (blue), respectively. Horizontal gray lines indicate thresholds for a significant proportion of neurons (binomial test, p<0.05 without or with multiple-comparisons correction, light and dark gray lines, respectively). Light and dark color-filled bars are those for which the threshold was crossed without or with multiple-comparisons correction. Right, sessions without significant behavioral evidence of proactive inhibition do not show this firing rate difference between conditions (same format as left panels; n=173 neurons). (**B**) Two individual example neurons demonstrating the proactive elevation of firing rate before the *Go!* cue. Top: averaged firing rates in each condition. Shaded band, ± SEM across trials. Bottom: raster plots of individual trials. Trials are sorted by reaction times (RTs). (**C**) Neurons were categorized as decrease-type or increase-type, based on an 'Initiation Score'. We defined the 'Initiation Score' for each neuron as the change in (Z-scored) firing rate in the 0.2 s before Center Out (inset shows example neuron). Plots show average firing of each subpopulation (top; ± SEM) and individual cells (normalized average firing, subtracting 'baseline' firing at 0.2 s before Center Out) sorted by Initiation Score (bottom). (**D**) No relation between Initiation Score and proactive inhibition (assessed as the difference between Maybe-Stop-Contra and No-Stop trials, in the 200 ms before *Go!* cue). Bar graph (inset) shows that on average, both increase- and decrease-type neurons modestly increase firing with proactive inhibition (Wilcoxon signed rank tests in each group). Error bar is ± SEM across neurons. (**E**) We defined the 'Selection Score' for each neuron as the integral of the difference in (Z-scored) firing rate between Contra and Ipsi actions during the 0.2 s epoch before Center Out. Remainder of panel is as C, but for Selection Score (individual cell plots show normalized firing rate for Contra minus Ipsi actions). (**F**) Significant negative correlation

*Figure 2 continued on next page*

*Figure 2 continued*

between Selection Score and proactive inhibition. Bar graph (inset) shows that Ipsi>Contra neurons preferentially increase activity on Maybe-Stop-Contra trials. (**G**) Same result as F, using the format of panel A to illustrate time course.

The online version of this article includes the following figure supplement(s) for figure 2:

**Figure supplement 1.** Locations of recorded cells.

**Figure supplement 2.** Ipsi>Contra type cells.

**Figure supplement 3.** Normalized firing rates.

ms relative to Center Out (disregarding the direction of movement). The Selection Axis connects the midpoints of the Contra and Ipsi trajectories (averaging across the same time epoch).

When we projected Maybe-Stop-Contra data onto these axes, we found that overall SNr population activity showed a clear shift before the *Go!* cue (*Figure 3A*, middle). This shift was especially pronounced on the Selection Axis, with a highly significant bias toward Ipsi (*Figure 3A*, bottom). There was also a significant bias on the Initiation Axis (toward movement). These two biases were associated with distinct functional cell classes (*Figure 3B and C*). Ipsi>Contra cells showed a strong Selection Axis bias, without a significant Initiation Axis bias (*Figure 3B*), and the converse was seen for Contra>Ipsi cells (*Figure 3C*). This finding provides further evidence that Ipsi>Contra cells generate an Ipsi bias during selective proactive inhibition.

Moreover, the specific neural state of preparation was related to the subsequent specific behavioral outcome of the trial. Trajectories during wrong choice trials were biased both toward Ipsi action and toward initiation at the time of *Go!* cue (*Figure 3—figure supplement 1D*), consistent with our prior GPe results (*Gu et al., 2020*). Trials in which rats failed to initiate actions (limited hold violations) did not show any statistically significant bias at the *Go!* cue time, although after the *Go!* cue the neural trajectory showed movement in the direction away from action initiation (*Figure 3—figure supplement 1D*).

## Less regular firing with proactive inhibition

Movement slowing in PD has been associated with changes in basal ganglia firing patterns (*Sharott et al., 2014*; *Tai, 2022*). We therefore examined whether the slower movement initiation during proactive inhibition is similarly accompanied by changes in SNr firing patterns. For each neuron we calculated the coefficient of variation (CV) of inter-spike intervals, and the proportion of spikes within bursts (*Figure 4A and B*; using the Poisson surprise method; *Legéndy and Salcman, 1985*). To obtain sufficient numbers of inter-spike intervals, both analyses used longer (3 s) epochs before the *Go!* cue.

Proactive slowing was associated with altered spiking variability, in several distinct analyses. Neurons recorded in sessions with significant proactive inhibition behavior (n=446) showed more irregular and bursty firing compared to neurons (n=173) in sessions that did not (*Figure 4C*). The degree of irregularity was correlated with the mean session-wide RTs (*Figure 4—figure supplement 1A*). Increased irregularity and bursting were also seen during Maybe-Stop-Contra trials compared to No-Stop trials; this occurred selectively in those sessions with a significant proactive inhibition effect (*Figure 4D*, left), and not those without (*Figure 4D*, right). Moreover, the degree of increased irregularity between conditions was correlated with the magnitude of the proactive effect on RTs on each session (*Figure 4—figure supplement 1B*). Together, these results show that engaging proactive inhibition increases spiking variability both within- and between-sessions.

## Proactive inhibition increases the variability of neural trajectories

How could increased spiking variability contribute to the slowing of RTs? At the population level, spike variability would correspond to more erratic state-space trajectories before the *Go!* cue (*Figure 5A*). This would result in a more variable state at the (unpredictable) time of *Go!* cue onset. If effective movement preparation involves positioning neural activity within a 'optimal subspace', as previously proposed (*Churchland et al., 2006*), this more variable state would in turn result in RTs that are longer (on average) and more variable (across trials).

To assess this idea we quantified trajectory fluctuations on individual trials (*Figure 5B–D*). We included sessions (n=27) in which more than five neurons were recorded simultaneously, and examined trajectories in the 3 s window before the *Go!* cue (for comparison to our CV measure). In the

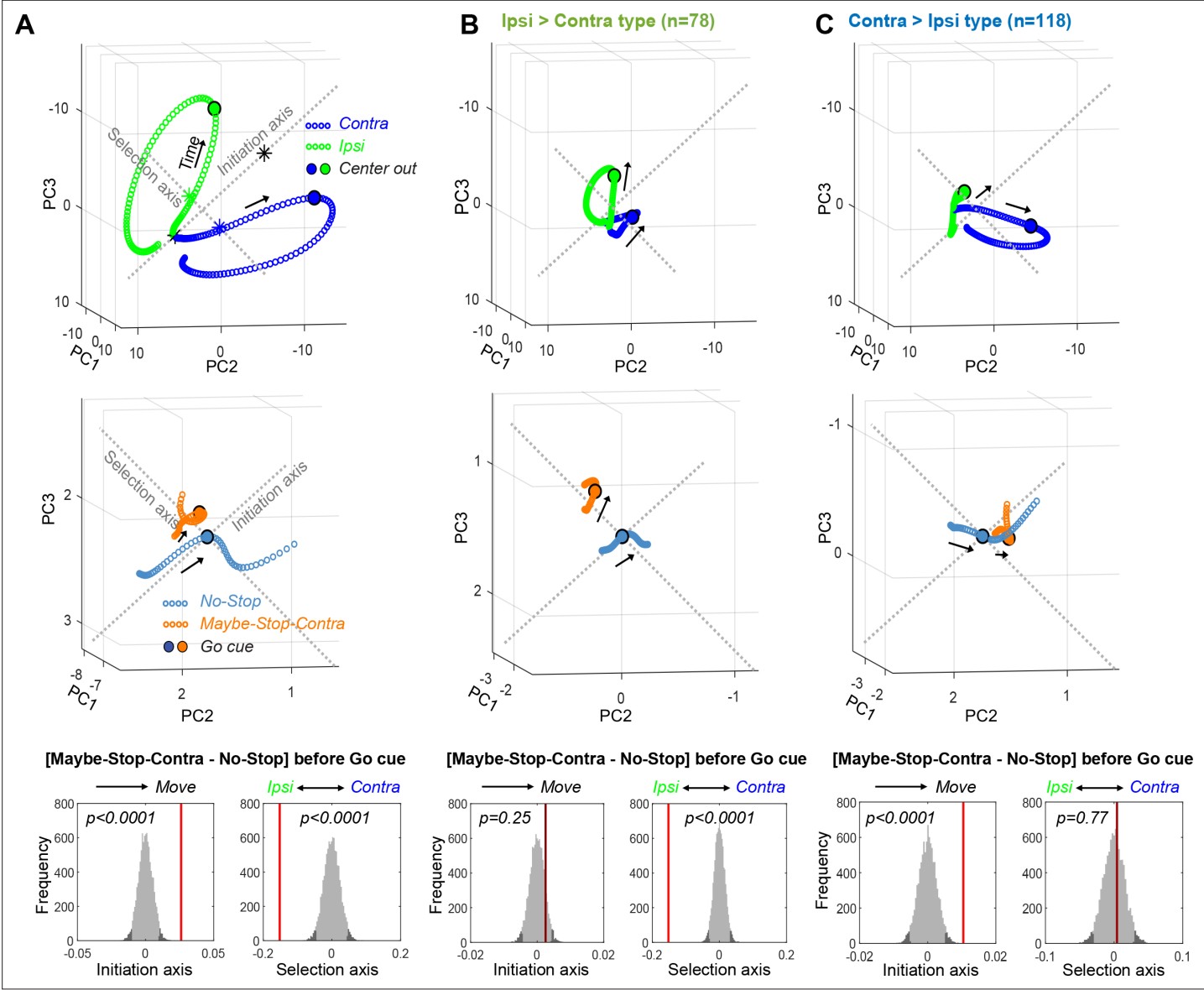

**Figure 3.** Biased substantia nigra pars reticulata (SNr) population dynamics during proactive inhibition. (**A**) Top, Overall SNr state-space trajectories before and during Contra (blue) and Ipsi movements (green), in the state space of the first three principal components (PCs). Trajectories show ±250 ms around detected movement onset (Center Out, larger circles), with each small circle separated by 4 ms. 'Initiation Axis' joins the positions (black asterisks) 200 ms before and at action initiation (averaging Contra and Ipsi actions). 'Selection Axis' joins the means of each trajectory in the same epoch (colored asterisks). Middle, Comparing Maybe-Stop-Contra (orange) and No-Stop (blue) trials (±100 ms around *Go!* cue) in the same state space as above. Overall population state is visibly biased toward Ipsi along the Selection Axis. Bottom, Permutation tests of bias along each axis (average during –200 to 0 ms relative to *Go!* cue), using all 10 PCs. Red bars, observed results; gray, distributions of surrogate data from 10,000 random shuffles of trial-type labels. Dark gray indicates 5% of distributions at each tail. (**B**) As A, but for Ipsi>Contra cells only. The proactive bias toward Ipsi along the Selection Axis is more clearly visible. For comparisons, the PC dimension scale and Initiation/Selection axis are matched to the graph in A. (**C**) As A–B, but for Contra>Ipsi cells. These do not show a proactive bias on the Selection Axis, but on the Initiation Axis instead.

The online version of this article includes the following figure supplement(s) for figure 3:

**Figure supplement 1.** Principal component analysis (PCA).

same way as CV measures variability over time of an individual neuron (one dimension) compared to its mean rate, we can measure the within-trial variability of a population (n dimensions, without principal component analysis [PCA]) by comparing the Euclidean distance of each time point to the mean position (***Figure 5C***).

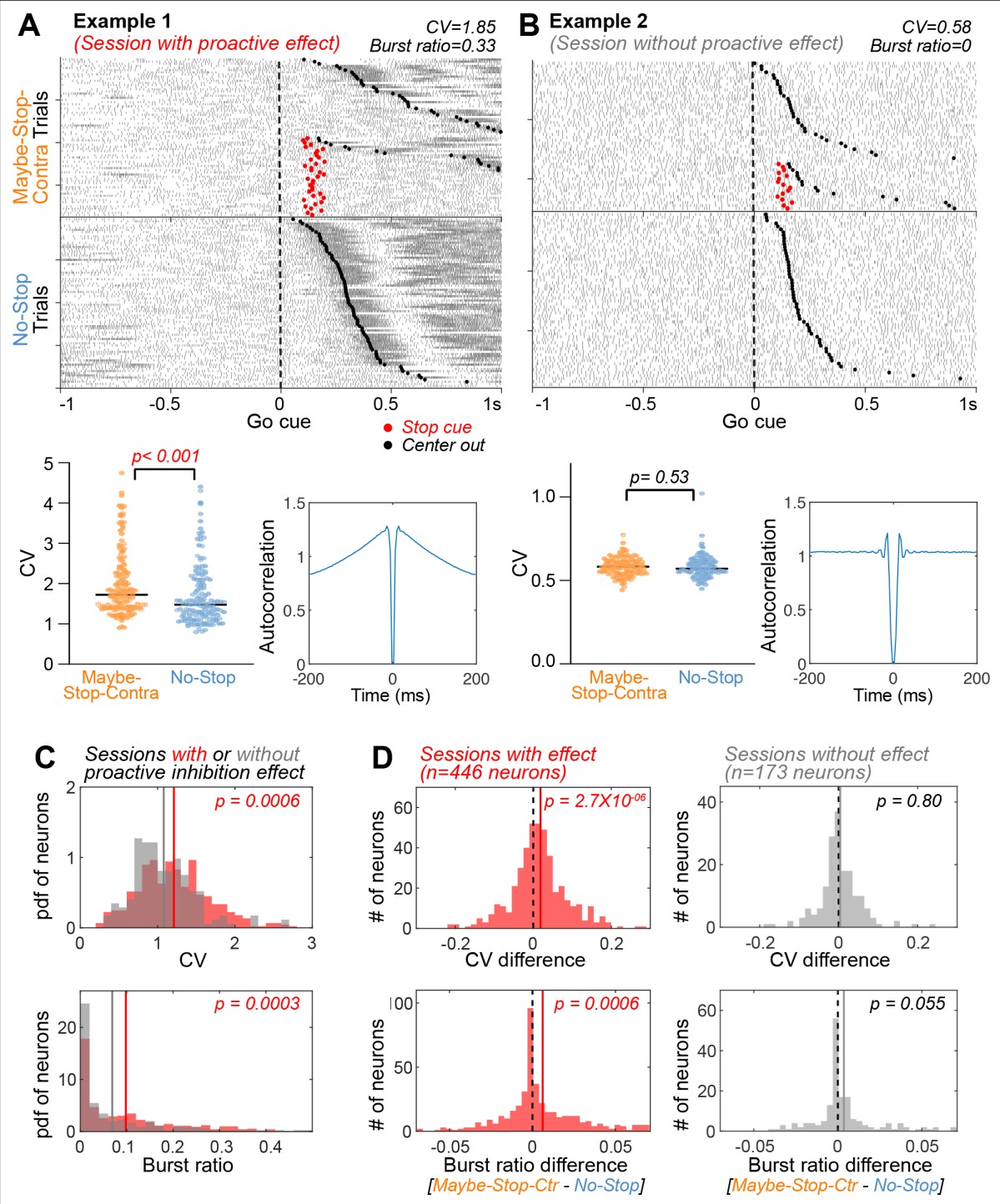

**Figure 4.** Substantia nigra pars reticulata (SNr) firing is more irregular and bursty with proactive inhibition. (**A**) An example neuron from a session with proactive inhibition effect, showing spike rasters (top; aligned on contra *Go!* cues, sorted by reaction times [RTs]), CVs of individual trials (bottom left, for the 3 s preceding Go! cues, Wilcoxon rank sum test) and autocorrelograms (bottom right; for the 3 s preceding *Go!* cues). CV: coefficient of variation of inter-spike intervals. (**B**) An example neuron from a session without a proactive inhibition effect. Same format as in (**A**). (**C**) Elevated CV and burst ratio of neurons recorded in sessions with behavioral evidence of proactive inhibition, compared to sessions without (Wilcoxon rank sum tests). This effect was also seen at the level of individual rats (***Figure 4—figure supplement 1C***). Gray and red lines indicate mean of sessions with and without proactive inhibition effect, respectively. (**D**) Within-session comparison of Maybe-Stop-Contra and No-Stop trials shows increased CV and burst ratio when proactive inhibition is engaged (left). This effect is not present on sessions without significant proactive slowing of reaction times (right). Comparisons

*Figure 4 continued on next page*

*Figure 4 continued*

within each functional cell type and within each individual are shown in *Figure 4—figure supplements 2 and 3*. Wilcoxon signed rank tests. Dotted black lines indicates zero and colored lines indicate mean of the neurons.

The online version of this article includes the following figure supplement(s) for figure 4:

**Figure supplement 1.** Coefficient of variation (CV) in individual sessions and rats.

**Figure supplement 2.** Spiking varibilities of response selective cells.

**Figure supplement 3.** Individual rat data.

As expected, there was a strong relationship between CV of individual neurons and their corresponding population trajectory variability (*Figure 5E*). Furthermore, trajectory variability was significantly increased by proactive inhibition (*Figure 5F*). Trajectory variability increased during the hold period (examining the 0.8 s epoch just before the *Go!* cue, excluding trials with hold duration <0.8 s, Wilcoxon signed rank test, p=0.003), and was also observed even beforehand (0.8 s epoch before Nose In, Wilcoxon signed rank test, p=0.003). This increase in within-trial trajectory variability with proactive inhibition indeed resulted in a more variable state-space position at the time of *Go!* cue onset, across trials (*Figure 5G*). Consistent with our hypothesis, this variability of state-space position across trials was correlated with RT (*Figure 5H*).

## Proactive modulation of firing rate, and variability, is dissociable

Changes in preparation to move or stop can be evoked by explicit cues, as in our task design, but also by the subject's ongoing experience – notably, what happened on the previous trial (*Bissett and Logan, 2011*; *Pouget et al., 2011*). We therefore examined how such ongoing experience affects the proactive influences over SNr firing. Regardless of previous trial-type Maybe-Stop-Contra trials had slower RT (*Figure 6*, top row), and an Ipsi-biased SNr state-space position (*Figure 6*, third row). By contrast, the increased CV in SNr spiking was particularly apparent following Stop-fail trials (*Figure 6*, bottom). This suggests that increased spike variability occurs when rats are especially concerned to avoid hasty responses, having just failed to sufficiently restrain behavior on the previous trial. To further assess whether increased SNr variability is an adaptive response to excessive haste, we examined the CV of trials following other types of errors (*Figure 6—figure supplement 1*). Consistent with the hypothesis, CV significantly increased after premature responses (Center-out before *Go!* cue), but not after other types of error trials (Wrong choices, and failures to respond within the RT limit).

## Discussion

Our capacity to inhibit actions and thoughts can be influenced by a wide range of factors (*Bari and Robbins, 2013*). These include external cues (such as warning stimuli) and 'internal' processes such as attention and motivation (*Meyer and Bucci, 2016*). Our experimental design provides some useful constraints on which factors are relevant to our results. We used a standard, operational definition of selective proactive inhibition: slowing of RTs for the particular movement that *might* need to be cancelled. Since this behavioral effect was observed even when no *Stop!* cue was actually presented, and was direction-selective, it cannot be explained simply by (for example) priming of the more global, cue-evoked Stop mechanisms that support reactive inhibition (i.e. 'preparation to stop'). Instead, it implies an altered state already present by the time of the *Go!* cue.

We found evidence for this altered state in two aspects of SNr spiking. A subpopulation of movement-selective SNr neurons showed elevated firing before the *Go!* cue, and this was associated with a shift in population activity away from the specific restrained action and toward the alternative. At the same time, more erratic SNr firing results in a more variable state at the time of the *Go!* cue. This variability is associated with slowed RTs, plausibly because effective movement preparation involves achieving a more constrained range of network activity. Moreover, these firing rate and variability modulations reflect two dissociable mechanisms for proactive control, as they were recruited differently depending on events on the prior trial.

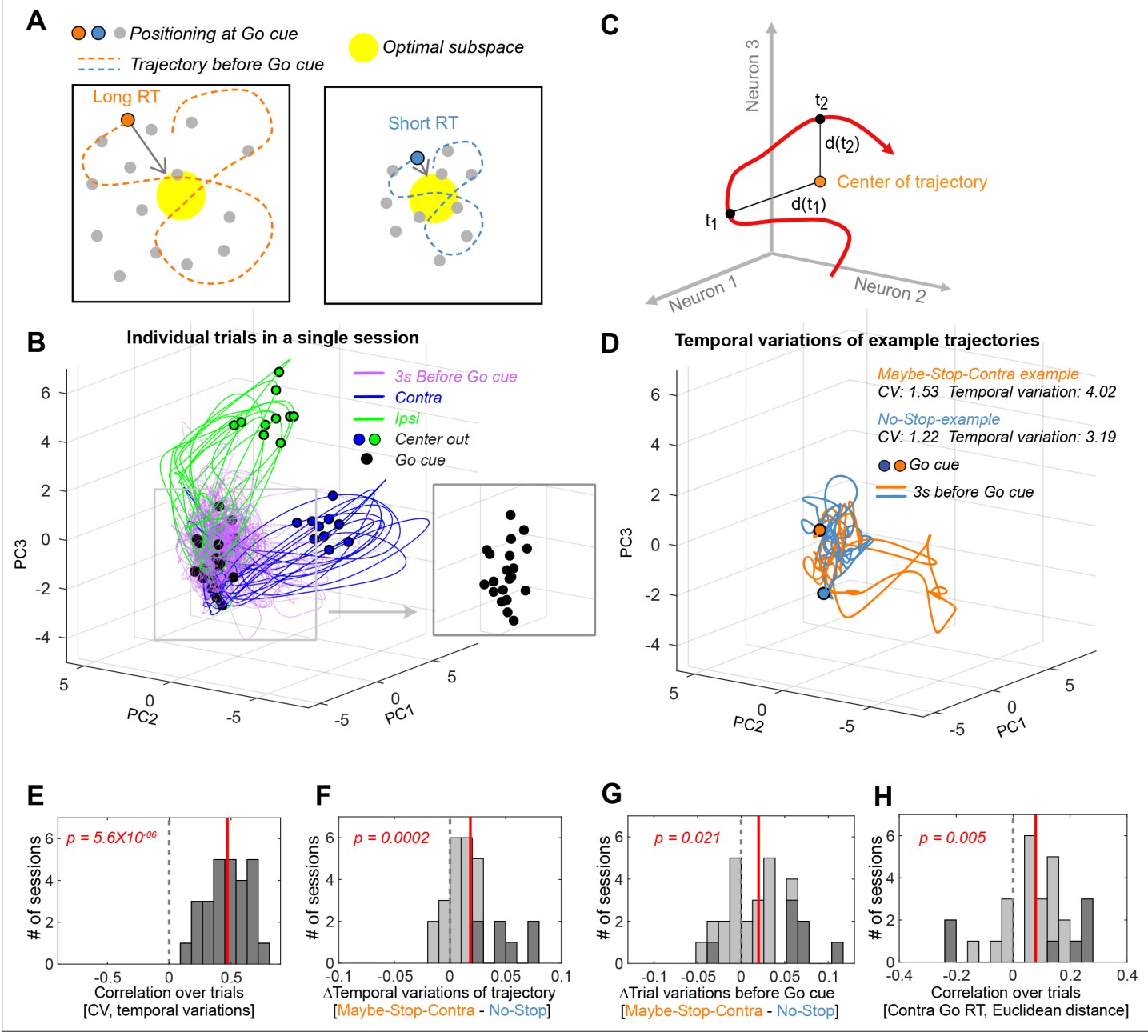

**Figure 5.** Altered variability of state trajectories with proactive inhibition. (**A**) Conceptual illustration for the relationship between trajectory variability and reaction times (RTs). Larger fluctuations in trajectory will tend to result in a position further away from the 'optimal subspace' when the *Go!* cue arrives. (**B**) Individual trial trajectories (10 trials each for Contra and Ipsi movements) for one example session (n=46 neurons). Trajectories are shown after principal component analysis (PCA) for visualization. (**C**) Trajectory variability was defined as the mean of the Euclidean distances at each time point to the mean position over the trajectory (in the full neural state space, without PCA). (**D**) Example trajectories for Maybe-Stop-Contra (orange) and No-Stop (blue) trials, before the *Go!* cue. Trajectories are shown after PCA for visualization. (**E**) Correlations between coefficient of variation (CV) and trajectory variability for each session. (**F**) Trajectory variability increased on Maybe-Stop-Contra, compared to No-Stop trials. (**G**) Across trials, the state-space position at *Go!* cue (–200 to 0 ms) was more variable for Maybe-Stop-Contra, compared to No-Stop trials. (**H**) Variability across trials of the state-space position at Go! cue (–200 to 0 ms) was positively correlated with RT. Trials with less than 100 ms reaction times were excluded because they would have already initiated the movement trajectory. For (**E**–**H**), sessions with more than five neurons (# of session = 27) were used for analysis (Wilcoxon signed rank test across session values). Dotted gray line indicates zero and red line indicates mean of the session values. Dark gray bars indicate sessions showing significant correlation (p<0.05 for (**E**), (**H**)), or conditional differences (Wilcoxon rank sum test, p<0.05 for (**F**), (**G**)).

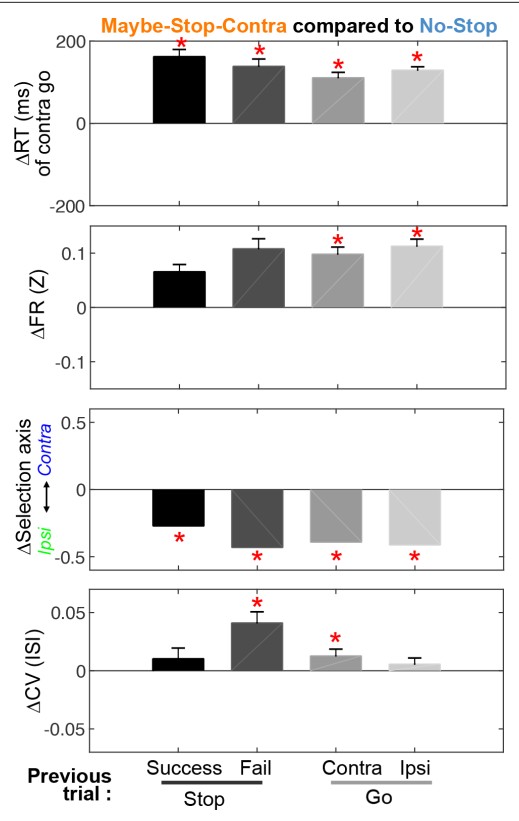

**Figure 6.** Feedback effect by previous trials and relation to trial outcomes. Differences in reaction time, firing rates, selection axis positions, and coefficients of variation (CVs) of Maybe-Stop-Contra trials, compared to the No-stop condition (zero), and separated by previous trial type. The effect of previous trial type reached significance for the difference in CV (Friedman's test, $X^2(3)=13.38$, $p=0.004$), but not for contra go RT ($X^2(3)=2.84$, $p=0.42$) nor in firing rates ($X^2(3)=5.09$, $p=0.17$). Firing rates and selection axis measures use Ipsi>Contra cells (during 200 ms before *Go!* cue); all cells are included in the CV calculation. *$p<0.05$ with Bonferroni multiple comparison correction (Compared to No-Stop trials, permutation test as in *Figure 3* for Selection axis, and Wilcoxon signed rank test for others). Error bars are ± SEM across sessions (n=60) for reaction times and across neurons (n=446) for firing rates and CV.

The online version of this article includes the following figure supplement(s) for figure 6:

**Figure supplement 1.** Post error trials.

## Basal ganglia dynamics and the nature of restraint

Our results add to prior evidence that, beyond simple sensory or motor correlates, SNr firing is modulated by internal factors such as task context (*Hikosaka and Wurtz, 1983a*; *Lintz and Felsen, 2016*). At the same time our operational definition of proactive inhibition does not fully constrain which internal factors are at play – and these may vary both within, and between, individual subjects. Basal ganglia activity is especially affected by changes in reward expectation (*Lauwereyns et al., 2002*) including in SNr (*Sato and Hikosaka, 2002*; *Yasuda and Hikosaka, 2017*), producing faster RTs toward more rewarding movements. In our proactive task the Maybe-Stop direction receives rewards at a lower rate (due to failures-to-Stop), so asymmetrical reward expectation might be at least partially responsible for the neural and choice bias toward Ipsi movements on Maybe-Stop-Contra trials. Moreover, this bias could have produced pre-selection of the Ipsi movement, a factor also known to modulate SNr activity (*Basso and Wurtz, 2002*). However, the slower Contra RTs on Maybe-Stop-Contra trials were not accompanied by faster Ipsi RTs, suggesting that slowing of Contra movements did not result simply from a greater preparation of Ipsi movements.

Another internal factor potentially at play is uncertainty: in the Maybe-Stop conditions there are more potential future action paths than under No-Stop. This could increase vacillation between different preparatory states as rats wait for the Go! cue (*Kaufman et al., 2015*; *Rich and Wallis, 2016*), with corresponding increases in neural variability.

The SNr state-space bias during proactive inhibition may arise from the 'indirect pathway' projection from GPe, which we found to have a similar neural bias toward Ipsi movements in a prior study (*Gu et al., 2020*). However, proactive inhibition also caused GPe population activity to shift further away from action initiation, which we did not observe for SNr. Furthermore, proactive inhibition produced no overall change in GPe firing rates, and we were not able to identify a distinct subpopulation of modulated GPe neurons. The reasons for these GPe: SNr differences are not clear. Recent modeling has demonstrated that the influence of the GABAergic GPe inputs over SNr neurons can be far more complex (*Simmons et al., 2020*) than shown in classic rate models of basal ganglia function, for example, switching from inhibitory to excitatory in an activity-dependent manner (*Phillips et al., 2020*). Alternatively, the shift in SNr Ipsi>Contra neuron firing with proactive inhibition may reflect other basal ganglia inputs, particularly the direct pathway input from striatum, or hyperdirect via the STN (*Schmidt et al., 2013*).

## Impact on downstream targets

How might the elevated activity of SNr Ipsi>Contra cells help restrain actions? To some extent, this elevation is compatible with classic models in which pauses in basal ganglia outflow disinhibit downstream targets to evoke movement (*Chevalier and Deniau, 1990*). Increased SNr activity while waiting for a Go! cue would enhance inhibition of movements, and the release of this inhibition might require longer to achieve, extending RTs. However, many of the Ipsi>Contra cells with elevated firing in the Maybe-Stop condition did not have movement-related pauses, but rather increases (*Figure 2—figure supplement 2A*). Such increasing-type SNr cells have been commonly reported (*Bryden et al., 2011*; *Joshua et al., 2009*; *Lintz and Felsen, 2016*), but their functional roles remain unclear. They may help suppress competing, alternative actions (*Gulley et al., 2002*), by acting either on downstream targets or through collateral inhibition of other SNr cells (*Brown et al., 2014*). Further studies, likely incorporating additional techniques, will be needed to resolve these and other possibilities.

## Behavioral control and neural variability

Our study also leaves unresolved the origins of the increased SNr spike variability we observed with proactive inhibition. Like GPe and STN neurons, SNr neurons are intrinsic pacemakers that spontaneously fire regularly even if their inputs are blocked (*Zhou and Lee, 2011*). These inputs – which include the aforementioned direct and indirect pathways, and also extensive local collaterals within SNr (*Brown et al., 2014*; *Mailly et al., 2003*) – thus alter SNr spiking by either accelerating or delaying the occurrence of the next spike. Changes in spike time variability presumably reflect either changes in local network properties, for example, due to neuromodulation (*Delaville et al., 2012*) or changes in the statistics of extrinsic inputs. Causal manipulation of specific incoming synapse sets within SNr (e.g. optogenetic suppression of GPe or STN inputs) might usefully reveal the origins of neural variability and establish its contribution to proactive inhibition.

More variable SNr inter-spike intervals within-trials were associated with more variable trajectories through a more variable network state at *Go!* cue, and longer and more variable RTs. In the cortex, decreases in spike variability have been previously linked to various processes including stimulus onset (*Churchland et al., 2010*), attention (*Cohen and Maunsell, 2009*), and movement preparation (*Churchland et al., 2006*). There is also evidence of within-trial changes in spike variability in the striatum (*Berke, 2011*). To our knowledge, there are no prior observations of specific task cues evoking *increases* in variability between or within trials. Our finding that variability increases with proactive inhibition – in particular following Stop-fail or premature trials – suggests that variability might be actively elevated as part of a behavioral strategy. This would fit with proposals that neural variability can confer behavioral advantages, such as increased flexibility (*Waschke et al., 2021*). Alternatively, increased neural variability with proactive inhibition may simply reflect the *absence* of a reduction in variability that accompanies movement preparation. In this way, preparation to stop would consist, at least in part, of less preparation to go. Either way, our results demonstrate that cognitive control strategies can operate through shifts in neural variability.

Connecting single-neuron measures of variability to state-space concepts of movement preparation offers an intriguing perspective on PD, which is characterized by both slowed movements (*Low et al., 2002*) and elevated single-neuron spike variability (*Dorval et al., 2008*; *Lobb, 2014*; *Willard et al., 2019*). A more variable state of preparation may help explain why *average* RTs and movement times are slowed in PD, yet the fastest movements are still preserved (*Mazzoni et al., 2007*). An important goal for future studies is to better understand how shifts in neural variability occur in the basal ganglia, and whether they originate from the same mechanisms during both proactive inhibition and PD.

# Materials and methods
## Animals

All animal experiments were approved by the University of California, San Francisco Committee for the Use and Care of Animals (#AN181071). Adult male Long-Evans rats were housed on a 12 hr/12 hr reverse light-dark cycle, with training and testing performed during the dark phase.

## Behavior

The rat proactive stop task has been previously described in detail (*Gu et al., 2020*). Briefly, rats were trained in an operant chamber (Med Associates, Fairfax, VT) which had five nose-poke holes on one wall, a food dispenser on the opposite wall, and a speaker located above the food port. Each trial starts with one of the three starting ports illuminated to indicate the *Stop!* cue probabilities ('Light On'), and the same start port was repeated for 10–15 trials. Inter-trial intervals were randomly selected between 5 and 7 s. The mapping of stop probabilities to nose-poke locations are counterbalanced between rats but maintained in each rat (*Table 1*).

## Electrophysiology

We recorded SNr data from 10 rats (all animals in which we successfully recorded SNr neurons during contraversive proactive inhibition). Each rat was implanted with 15 or 30 tetrodes in bundles bilaterally targeting SNr (*Figure 2—figure supplement 1*). Wide-band (0.1–9000 Hz) electrophysiological data were recorded with a sampling rate of 30,000/s using an Intan RHD2000 recording system (Intan Technologies). All signals were initially referenced to a skull screw (tip-flattened) on the midline 1 mm posterior to lambda. For spike detection we re-referenced to an electrode common average, and wavelet-filtered (*Wiltschko et al., 2008*) before thresholding. For spike sorting we performed automatic clustering units using MountainSort *Chung et al., 2017* followed by manual curation of clusters. Approximately every two to three sessions, screws were turned to lower tetrodes by 100–160 μm; to avoid duplicate neurons, we did not include units from the same tetrode from multiple sessions unless the tetrodes had been moved between those sessions. We further excluded a small number of neurons on the same tetrode that appeared to be potential duplicates based on waveforms and firing properties (e.g. firing rates, CV, and behavioral correlations), even though the tetrode had been moved. After recording was complete, we anesthetized rats and made small marker lesions by applying 5–10 μA current for 20 s on one or two wires of each tetrode. After perfusing the rats, tissue sections (at 40 μm) were stained with cresyl violet or with CD11b antibody and compared to the nearest atlas section (*Paxinos and Watson, 2006*).

## Data analysis

### Firing rates and functional classification of neurons

Firing rates were smoothed using a Gaussian kernel (30 ms SD) and normalized (Z-scored) using each neuron's session-wide mean and SD. Most analyses were done using these normalized firing rates, except the fraction of units (*Figure 2A and G*, bottom, using binned firing rates) and bursting analysis (*Figure 4*, see below).

Units were categorized into increase or decrease types using the 'Initiation score'. For example, if the firing at movement onset was significantly increased compared to the 200 ms before movement onset (Wilcoxon signed rank test, p<0.05; including all successful go trials in the No-Stop condition), the cell is defined as 'increase type'. The categorization into Contra>Ipsi or Ipsi>Contra types was done using the 'Selection score'; for example, if the mean firing rates during the 200 ms preceding movement onset was significantly bigger for the Ipsiversive compared to Contraversive movement during the No-Stop condition (Wilcoxon signed rank test, p<0.05), the cell was defined as 'Ipsi>Contra type'.

### Bursting

Spike bursts were detected using the Poisson surprise method (*Legéndy and Salcman, 1985*) with a surprise threshold of 5, and the burst ratio was calculated as the number of spikes fired in bursts divided by the number of all spikes in each unit. The CV of inter-spike intervals and burst ratio were calculated in each trial using a 3 s time window before the *Go!* cue, and averaged across all trials (for sessions with/without proactive inhibition comparison) or across each trial type (for conditional differences).

PCA was done largely as previously described (*Gu et al., 2020*). The smoothed, normalized average time series for Contra and Ipsi actions (500 ms each, around Center Out, *Figure 3—figure supplement 1A*) were used for PCA. This population activity matrix **R** is zero-centered, and after using MATALB 'svd' function, the PC scores (S) was calculated as S=**R**W, where W is the right singular vectors.

To examine proactive inhibition effect at the level of the whole SNr population, we defined 'Selection' and 'Initiation' axes using activity during Contra and Ipsi actions. To match the analysis to the Initiation and Selection scores (*Figure 2*), we defined the Initiation Axis by connecting state-space points 200 ms before and at action initiation (averaging Contra and Ipsi trajectories together), and the Selection Axis by connecting the mean of Contra trajectories to the mean of Ipsi trajectories (again using the epoch from –200 to 0 ms relative to action initiation). The projections onto the Initiation and Selection Axis were calculated as the dot product of the state-space position vector and the axis vector.

For mapping of before *Go!*cue time series data (**R′**) into the Contra and Ipsi action related PC dimension, new PC scores were calculated by S'=**R′**W after matching the zero values of **R′** to the matrix **R**. For subpopulation analysis (*Figure 3B, C*), selected unit's population vector (e.g. **R**$_{(ipsi)}$) and the right singular vectors (e.g. W$_{(ipsi)}$) were used to calculate the selective unit's PC scores (S$_{(ipsi)}$=**R**$_{(ipsi)}$ W$_{(ipsi)}$).

## Permutation tests

To test if conditional differences are statistically significant, we ran permutation tests by randomly shuffling the two comparing trial conditions for each neuron (10,000 shuffles). The original distance between the two comparing conditions were compared to the distance between shuffled trial conditions after projecting the distances onto the Initiation or Selection axis.

## Variability of neural trajectories

To examine how the increased variability of individual neurons affects population dynamics, we calculated the temporal variability of neural state-space trajectories. We included data from 3 s time windows before the *Go!* cue, in each session (n=27) with at least 5 recorded units and a significant proactive inhibition effect. Trajectory variability V corresponds to the mean Euclidean distance between each time point of the trajectory and the center of the trajectory, that is, how much the trajectory wanders around. For this analysis, we did not apply PCA to reduce dimensionality. Also, the trajectory variability was normalized by the number of neurons in each session. Thus, for example, if the normalized FR of population neurons (N) at time (t) is defined as a vector $\vec{R}_t$ , and the center of trajectory (mean FR across t=1,…,T, where T=1500 for 3 s time window with 500 Hz sampling rate) is defined as a vector $\vec{R}_c$. Variability (V) of a neural trajectory is defined as:

$$V = \frac{1}{T\sqrt{N}} \sum_{t=1}^{T} \| \vec{R}_t - \vec{R}_c \|$$

## Acknowledgements

We thank Charles Wilson and members of the Berke Lab for valuable feedback.

## Additional information

### Funding

| Funder | Grant reference number | Author |
| --- | --- | --- |
| National Institute of Mental Health | R01MH101697 | Joshua D Berke |
| National Institute of Neurological Disorders and Stroke | R01NS123516 | Joshua D Berke |
| CHDI Foundation | A-13733 | Joshua D Berke |

The funders had no role in study design, data collection and interpretation, or the decision to submit the work for publication.

## Author contributions
Bon-Mi Gu, Conceptualization, Data curation, Software, Formal analysis, Investigation, Visualization, Methodology, Writing - original draft; Joshua D Berke, Conceptualization, Resources, Supervision, Funding acquisition, Writing - original draft, Writing - review and editing

## Author ORCIDs
Bon-Mi Gu 🔟 http://orcid.org/0000-0002-3060-0803
Joshua D Berke 🔟 http://orcid.org/0000-0003-1436-6823

## Ethics
All animal experiments were approved by the University of California, San Francisco Committee for the Use and Care of Animals (#AN181071).

## Decision letter and Author response
Decision letter https://doi.org/10.7554/eLife.82143.sa1
Author response https://doi.org/10.7554/eLife.82143.sa2

---

# Additional files

## Supplementary files
• MDAR checklist

## Data availability
Electrophysiology data with behaviors and the codes used for the analysis are available at figshare (https://figshare.com/, https://doi.org/10.6084/m9.figshare.20409858).

The following dataset was generated:

| Author(s) | Year | Dataset title | Dataset URL | Database and Identifier |
|---|---|---|---|---|
| Bon-Mi G, Berke JD | 2022 | Altered basal ganglia output during self-restraint | https://doi.org/10.6084/m9.figshare.20409858.v3 | figshare, 10.6084/m9.figshare.20409858.v3 |

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
