## [Editor Report]

The article provides an interesting and timely insight into the role of the basal ganglia output in proactive inhibition. By examining single-cell responses as well as population activity, the authors establish neural signals of behavioral control and show that animals' outcome history influences both firing rates and variability of basal ganglia output activity.

---

## [Decision Letter]

**Decision letter after peer review:**

Thank you for submitting your article "Altered basal ganglia output during self-restraint" for consideration by *eLife*. Your article has been reviewed by 2 peer reviewers, and the evaluation has been overseen by a Reviewing Editor and Joshua Gold as the Senior Editor. The following individual involved in the review of your submission has agreed to reveal their identity: Gidon Felsen (Reviewer #1).

Essential revisions:

Enthusiasm is high for this study which was viewed as important for advancing our understanding of the role of the basal ganglia in behavioral inhibition. Although no new experiments are requested, the manuscript would benefit from some revisions to increase its clarity and a clearer discussion of how these results fit into the broader field.

A list of essential revisions to improve the clarity and presentation of the work are:

1. An additional analysis where within-session performance is analyzed.

2. Figure 2C-E: Population heat maps should be separated for ipsi vs. contra movements.

3. Clarify whether data for "maybe stop" trials only include Go trials, or if it also includes (failed) stop trials, and consider comparing data for these two types of trials, if that was not already done.

4. The Discussion section should be expanded to include more discussion of prior literature concerning how SNr neurons respond during movement and how upstream and downstream nuclei might be affected.

5. The writing was too concise in some places, particularly for Figure 2 and 3. Expansion of the rationale and methods for the analytical approaches used in these figures will clarify the paper.

*Reviewer #1 (Recommendations for the authors):*

– In Figure 2C, E, it would be helpful to see the population heat maps separately for ipsilateral and contralateral movements (as they are shown in the summary plots). Given that so many SNr neurons are direction selective, it is difficult to interpret activity averaged over both directions.

*Reviewer #2 (Recommendations for the authors):*

I first recommend better describing and sometimes even motivating the methodologic choices of the present work. A significant part of the methods is described in figure legends. Sometimes the authors refer to a previous publication (Gu et al., 2020) instead of providing the information. This is fine to some extent, but I think that in the present case, including more information in the Methods section will help readers to better appreciate the authors' work. For example, it is unclear how "task-related" neurons have been categorized as such. It is also not clear to me what is the added value of the state-space analyses compared to the single-unit approach, especially those shown in Figure 3. As I understand it, the computation through the dynamics framework does not consider the "task-related" activity of cells as an important criterion. Instead, all cells are included to extract latent dynamics which, as a whole, are supposed to underlie behavior. I think that the two ways of analysing neural activity should be better motivated. Also, in the analysis shown in Figure 3, I wonder why trajectories of neurons recorded during sessions in which rats successfully applied proactive inhibition are not compared against trajectories depicted by neurons monitored during sessions in which rates failed to apply inhibition.

My second recommendation is to improve the Discussion section by commenting with more details on the points mentioned above: the unexpected modulation of SNr neurons given their pre-movement pattern; the comparison between GPe and SNr; the impact of SNr activity on target areas and, more globally, the broad picture of the role of the basal ganglia during movement inhibition control (both active and reactive).

---

## [Author Response]

Essential revisions:Enthusiasm is high for this study which was viewed as important for advancing our understanding of the role of the basal ganglia in behavioral inhibition. Although no new experiments are requested, the manuscript would benefit from some revisions to increase its clarity and a clearer discussion of how these results fit into the broader field.

We thank the Reviewers and Editors for their enthusiasm and thoughtful comments, which are all addressed below.

A list of essential revisions to improve the clarity and presentation of the work are:1. An additional analysis where within-session performance is analyzed.

To clarify, our analyses do include within-session comparisons (in particular, we show clear differences between Maybe-Stop and No-Stop trials within sessions). For further details and additional within-session analyses please see the response to Reviewer 1 below.

2. Figure 2C-E: Population heat maps should be separated for ipsi vs. contra movements.

We now include a version of these heat maps separated for ipsi vs contra movements (Figure 2-supp. 3).

3. Clarify whether data for "maybe stop" trials only include Go trials, or if it also includes (failed) stop trials, and consider comparing data for these two types of trials, if that was not already done.

We have added a clarifying line in Results:

“Since this time epoch [being analyzed] is before any Stop cue could occur, we included Maybe-Stop trials regardless of whether a Stop cue was subsequently presented or not”.

4. The Discussion section should be expanded to include more discussion of prior literature concerning how SNr neurons respond during movement and how upstream and downstream nuclei might be affected.

We have made a range of changes to the Discussion, incorporating additional prior literature.

5. The writing was too concise in some places, particularly for Figure 2 and 3. Expansion of the rationale and methods for the analytical approaches used in these figures will clarify the paper.

We have expanded the text in various places to clarify the rationale and methods for analyses.

Reviewer #1 (Recommendations for the authors):– In Figure 2C, E, it would be helpful to see the population heat maps separately for ipsilateral and contralateral movements (as they are shown in the summary plots). Given that so many SNr neurons are direction selective, it is difficult to interpret activity averaged over both directions.

We have now added these separate heat maps (Figure 2-supp. 3).

Reviewer #2 (Recommendations for the authors):I first recommend better describing and sometimes even motivating the methodologic choices of the present work. A significant part of the methods is described in figure legends. Sometimes the authors refer to a previous publication (Gu et al., 2020) instead of providing the information. This is fine to some extent, but I think that in the present case, including more information in the Methods section will help readers to better appreciate the authors' work. For example, it is unclear how "task-related" neurons have been categorized as such. It is also not clear to me what is the added value of the state-space analyses compared to the single-unit approach, especially those shown in Figure 3. As I understand it, the computation through the dynamics framework does not consider the "task-related" activity of cells as an important criterion. Instead, all cells are included to extract latent dynamics which, as a whole, are supposed to underlie behavior. I think that the two ways of analysing neural activity should be better motivated. Also, in the analysis shown in Figure 3, I wonder why trajectories of neurons recorded during sessions in which rats successfully applied proactive inhibition are not compared against trajectories depicted by neurons monitored during sessions in which rates failed to apply inhibition.My second recommendation is to improve the Discussion section by commenting with more details on the points mentioned above: the unexpected modulation of SNr neurons given their pre-movement pattern; the comparison between GPe and SNr; the impact of SNr activity on target areas and, more globally, the broad picture of the role of the basal ganglia during movement inhibition control (both active and reactive).

Thank you for these recommendations. We have made a range of changes to improve clarity, rationales for analyses, and the Discussion. This includes removing the confusing term “task-related activity”, by which we meant simply examining activity around task events. All distinct single neurons are included in all analyses.